# Comparing Methods to Collect and Geolocate Tweets in Great Britain

**Stephan Schlosser** [1,*], **Daniele Toninelli** [2] and **Michela Cameletti** [2]

1 Center of Methods in Social Sciences, University of Göttingen, 37073 Göttingen, Germany
2 Department of Economics, University of Bergamo, 24127 Bergamo, Italy; daniele.toninelli@unibg.it (D.T.); michela.cameletti@unibg.it (M.C.)
* Correspondence: stephan.schlosser@sowi.uni-goettingen.de

**Abstract:** In the era of Big Data, the Internet has become one of the main data sources: Data can be collected for relatively low costs and can be used for a wide range of purposes. To be able to timely support solid decisions in any field, it is essential to increase data production efficiency, data accuracy, and reliability. In this framework, our paper aims at identifying an optimized and flexible method to collect and, at the same time, geolocate social media information over a whole country. In particular, the target of this paper is to compare three alternative methods to collect data from the social media Twitter. This is achieved considering four main comparison criteria: Collection time, dataset size, pre-processing phase load, and geographic distribution. Our findings regarding Great Britain identify one of these methods as the best option, since it is able to collect both the highest number of tweets per hour and the highest percentage of unique tweets per hour. Furthermore, this method reduces the computational effort needed to pre-process the collected tweets (e.g., showing the lowest collection times and the lowest number of duplicates within the geographical areas) and enhances the territorial coverage (if compared to the population distribution). At the same time, the effort required to set up this method is feasible and less prone to the arbitrary decisions of the researcher.

**Keywords:** Twitter; geographical coverage; social media; big data; geolocation; spatial data collection

## 1. Introduction

We are living in an era characterized by a constant and massive production of a huge amount of data on a daily basis (or even at higher frequencies). In this world of "big data", research linked to data collection methods and data quality issues is becoming more and more important, since these aspects can have a relevant role in a lot of decision-making processes. In particular, data production efficiency and data accuracy and reliability are fundamental premises for taking timely and solid decisions. From this perspective, the Internet represents a very interesting new type of data source and research tool. In fact, traditional data collection methods (e.g., questionnaire-based surveys) can be implemented through the Internet [1]. In addition, a huge amount of data can be retrieved from social networks [2–4] or by means of web-scraping techniques [5]. However, can these data be used as a valid, truthful, and reliable support in making decisions or in supporting research in various fields? The balance between potentialities and risks linked to these new data still needs to be fully explored [6]. On the one hand, the web data collection has relevant advantages. For example, data can be automatically collected and stored for relatively low costs [7]. The data collection phase has almost no boundaries (i.e., we can easily reach any part of the globe) [8]. New types of data or indicators (including paradata, location or sensor information, digital data such as pictures, videos, and voice messages) can be collected or produced [9]. The data can be made available immediately, producing very large sized datasets that can be used for a wide range of purposes. For example, they can be employed for supporting theoretical and applied research projects and/or as evidence

for making timely decisions [10]. They also allow scientists to have constant and quick updates, with information available almost in real time.

On the other hand, the use of the Internet or of social media as a data source is linked to potential drawbacks, and poses challenges and risks of different types. For instance, data collected through social media are frequently unstructured and/or not collected for the specific purposes of the research; this also happens because there is a lack of control by the researcher, in comparison to other more traditional data collection methods (such as surveys). Moreover, researchers should carefully check the collection strategy in order to assess the data representativeness and potential sampling bias, as well as the possibility of integration with other sources (such as traditional surveys or official statistics data). In addition, once they are programmed, despite being automatic, data collection procedures require constant maintenance (i.e., revisions and updates). For example, in developing sentiment analyses, there could be the need of using updated dictionaries or of modifying automatic methods to clean and select in-scope data. Furthermore, changes of websites' content, structures, or links require a periodic monitoring and update of the downloading activities, in order to avoid the crash of web scraping processes. Finally, raw data collected from the web need to go through continuous, accurate, and time-consuming pre-processing phases, in order to organize, clean, adapt, and transform them into useful indicators and, at the end, into information. In addition, the large, potentially infinite, datasets that could be obtained from the web lead to computational challenges in terms of processing time and storage resources.

In order to exploit, as much as possible, the advantages and the opportunities offered by these new types of data sources (i.e., Internet and social media), it is first of all necessary to find efficient strategies of data collection. In this regard, the first main objective of this paper is to identify an optimized and flexible method to collect social media data. In particular, we evaluate the capability, the accuracy, and the efficiency of three alternative methods in retrieving tweets, that is, 280-characters messages sent through the social media Twitter. We based our research on tweets because Twitter is one of the most widespread and used social media worldwide. It is currently ranked by Alexa as the 11th most popular site (Rank based on the world global Internet traffic and engagement over the past 90 days https://www.alexa.com/topsites): 321 million active users (as of the end of 2018) publish 500 million tweets on Twitter per day (5787 tweets every second https://blog.hootsuite.com/twitter-statistics/). Moreover, in the last years "Twitter's data has been coveted by both computer and social scientists to better understand human behavior and dynamics" [11]. In fact, the information enclosed in these short messages can be used for a wide set of objectives (e.g., evaluation of the impact of policies or of advertising campaigns, longitudinal and/or spatial analysis, impact assessment of events and news, forecast of macro-scenarios, and so forth) in different fields (social sciences, economics, demography, and so on). In addition to this, a broad set of new techniques (e.g., machine learning, text mining, sentiment analysis) can be fruitfully applied to such data for several purposes. Moreover, research options are not just limited to the study of tweets content. Paradata—which describe the data creation process by including information about space, time, used device, and so forth—are automatically collected and may contribute in increasing the knowledge on the studied phenomena. They can also have an important role in measuring the quality of the collected data or even in defining better their content, the context in which they were produced or in justifying their use in specific research. For example, it is possible to perform spatial analyses (e.g., studying the spatial distribution across areas of the sentiment for a particular topic, of the perceived well-being) or geographical-based studies (e.g., about the spread of a certain disease, the evaluation of implemented policies and marketing campaigns, the daily flows of commuters between areas). In this research context, it is indispensable to work with data linked to reliable and precise geographical information. Thus, the quality of the data collection method can be evaluated also by means of its capability of extracting/estimating correct spatial information [12]. Unfortunately, the location information is sometimes hard to obtain, sometimes being

unavailable, scarce, and/or of low reliability. For example, for messages posted on Twitter, the geographic location (i.e., latitude and longitude) is available for only 1–2% of the messages (https://developer.twitter.com/en/docs/tutorials/tweet-geo-metadata.html). Consequently, after the data collection, it is necessary to develop methods for geolocating the text messages, i.e., for assigning a geographic location to each message.

In this framework, our second main objective is to automatically and fully geolocate the information we collect, allowing us to, at the same time, to personalize the level of geographical resolution. In particular, this paper focuses on the study of the entirety of Great Britain (GB); all three methods we propose are able to automatically geolocate all the collected tweets. This means that we can link each text message to a specific sub-area of GB. In this study, the level of geographical resolution is defined by the NUTS-1 (Nomenclature of Territorial Units for Statistics) level macro regions (https://ec.europa.eu/eurostat/web/nuts/background). Moreover, the methods we consider aim at covering as completely and precisely as possible these geographical sub-areas, reproducing, at the same time, the population distribution and minimizing overlaps.

This work is based on all tweets available for GB from 15th January 2019 to 15th February 2019 and collected, in parallel, through the three different methods. The final dataset (referred to as "raw data") includes a total number of 119,505,204 tweets.

Our findings show that one of the three methods results as the best option, because it is able to collect the highest amount (and the highest percentage) of unique Tweets per collection time unit. This method is also able to optimize the collection time as well as the pre-processing phase load. At the same time, the initial effort in setting the collection (in terms of number and size of circles) is within acceptable limits, and the method is less prone to the arbitrary decisions of the researcher.

Section 2 discusses the main past works at the base of and linked to this paper. Then, in Section 3, we introduce the paper's main research hypothesis and its added value. Section 4 describes how we collect our data, geolocating tweets, and the criteria we used to compare the three methods we aim at evaluating. Section 5 provides the detailed results of this evaluation. In Section 6, we summarize and discuss our main findings and, finally, Section 7 concludes the paper, highlighting the main limitations of our study as well as ideas for further research.

## 2. Literature Review

With the recent quick and global spread of the phenomenon of social media, we started observing a massive daily production of data (e.g., commercial messages, personal posts, opinions, and so on). Moreover, new potentially interesting types of digital data have become available (including pictures, videos, GPS coordinates, sensor data, and so forth). In this new data production perspective, the role of researchers is more limited in comparison to traditional research settings. On the one hand, it is not possible to interact with the data provider, nor to directly manage or control the data production process. On the other hand, researchers cannot even observe directly who is providing the data, in which context, and so forth. Moreover, most of the time, no direct information is available about the background or about the socio-demographic characteristics of the subjects, despite there having been some attempts to better define, for example, Twitter users [13,14].

Nonetheless, social media like Twitter currently provide an infinite data repository which could be used, potentially, for several purposes [4]. Consequently, such a type of real-time data source has become one prominent tool in order to support the development of a very wide range of studies and data-driven decisions in different contexts. Just to cite a few examples, social media data are currently used in several different domains, such as: Public health [15,16]; disaster management (for creating crisis maps, assessing damages and detecting help requests, for driving ongoing relief activities or providing necessary information to relatives of those involved, or simply and exclusively as a live-information channel) [10,17–22]; politics (for evaluating the implementation of new policies, laws, or interventions, or studying the public opinion; for forecasting election results or monitoring

election periods) [11,23–25]; language studies [26–28]; monitoring of socio-demographic phenomena [29], transportation [30], or real time events [31]; journalism (with an increasing usage of contents—video, images, sounds—that eyewitness events or news) [32]; marketing research [33]; sentiment analysis [27]. Social media analytics can also be used with a more commercial perspective, for example by business managers, in order to evaluate or to forecast the performance of specific products or services, or to obtain feedbacks from customers, to drive targeted marketing and advertising strategies, or to orientate production plans [34–36]. In addition, social media data could be experimentally used in the framework of open innovation [37], enlarging potential sources of external knowledge to accelerate firms' internal innovation. More specifically, these new types of data can be used to enrich studies focused on e-commerce companies, expanding open business model feedback loop platforms, such as the one of Alibaba [38]. Moreover, by means of a textual analysis of tweets regarding these companies and/or their products/activities, these data can support the new business models introduced by open innovation.

Generally speaking, the huge amount of data currently produced on a daily basis through social media can be seen as a limitless and constantly updated mine, in large part still unexplored. Consequently, researchers should become able to retrieve data and to explore and use this "big data" mine in order to obtain the information they need. In this framework, it becomes necessary to detect, study, and face the potential challenges and limits of such a data source. From this point of view, the challenges start in the phase of collection. First, because the available data are not produced and collected for the purposes they will be used for. Second, because they are rarely in a form that researchers can directly use [4]. In addition, collecting social media messages indiscriminately can cause issues, when it becomes necessary to filter or select posts that are linked to specific topics. The three main large-scale methods of data collection on social media platforms are introduced in Liang and Zhu [4] and include: "Log-data" collection, the use of Application Programming Interfaces (APIs), and the web scraping. Our work is based on the second approach and makes use of the Twitter API ( https://help.twitter.com/en/rules-and-policies/twitter-api). Lomborg and Bechmann [39] discuss the main opportunities and methodological issues linked with quantitative and qualitative research based on APIs, as well as the legal and ethical implications of using such tool for collecting data.

One of the most important usages of social media data is in analyses that take into account the spatial information. In such a framework, geolocation of tweets is crucial for many applications, such as real-time event detection, mobility studies, sentiment analysis, natural disaster analysis, and transportation planning [40]. In most of these studies, in order to achieve interesting and reliable results, one of the key aspects is the capability of associating reliable geodata information to the collected messages [12]. In fact, among the tweets available via the Twitter API, only 1–2% are geotagged, i.e., contain latitude–longitude coordinates [41–43]. Moreover, the geotag linked to social media content can be inaccurate: It can indicate either the location where the text was written or a place mentioned in the text. To make things worse, this information, when not missing, is frequently unreliable. In fact, Middleton et al. [42] p. 2 found that "geotags can be many kilometers away from where the subject matter is located". In addition to geotag information, some of the tweets may contain self-reported (mentioned) locations. Nevertheless, "the textual description of media posts can contain contextual location mentions that cannot be inferred from a geotag alone (e.g., 'Obama in Washington making a speech about China')" [42] p. 2. In addition, it was found that these cited locations are in most of the cases inaccurate or invalid (i.e., referring to non-existing locations) [44]. It is also possible for the user to specify a location in the Twitter account registration. However, this self-reported information can be out of date and not related to the location where tweets are actually sent. In this regard, Hecht et al. [45] p. 1 found that "34% of users did not provide real location information, frequently incorporating fake locations or sarcastic comments that can fool traditional geographic information tools". Given these problems, Middleton et al. [42] suggested

implementing location extraction techniques that can still add value to the information available from collected posts [12,44,46]. More generally, geocoding, geoparsing, or geotagging techniques can be used to deal with the problem of missing location in tweets [20,42]. What is in common between the three methods is that they all rely on the information extraction that takes place after the data retrieval. In particular, geocoding has the scope of "transforming a well-formed textual representation of an address into a valid spatial representation, such as a spatial coordinate or specific map reference" [42] p. 2. Geoparsing shares the same approach, but deals with unstructured free text (such as tweets): It starts by involving both location extraction (extracting location information from the text by using named entity recognition or named entity matching) and location disambiguation (i.e., the selection of the most likely locations among a set of possible location matches) before the final geocoding process. Geotagging uses statistical models and machine learning methods in order to assign spatial coordinates by analyzing the text content and/or the Twitter network (e.g., [40,44,47,48]).

Nowadays, there are several commercial geocoding services (for a review, see [42]) which deal with well-formatted location descriptions. Users can post a textual phrase and receive a likely location reference that matches it, including longitude and latitude spatial coordinates. Nevertheless, social media posts consist of informal and unstructured texts for which standard natural language processing tools do not perform well [46]. Moreover, the throughput of remote geocoding services is much lower than the real-time volumes of posts collected from social media (i.e., these services have rate limits).

The spread of research linked to the geolocating issue is a clear sign that current solutions are still affected by relevant limitations. Thus, how should we collect the huge amount of data produced by social media users? How can we handle the issue of missing geographical information that, in the case of Twitter, is characterizing most of the collected posts?

### 3. Goal, Research Hypothesis, and Added Value

In order to fill the research gap highlighted by the literature review, the main goal of our research is to find an efficient method for collecting tweets. In particular, our approach aims to collect data which are automatically geolocated at the desired area level (not at the point level), since the tweet location is a direct product of the collection methods. This feature allows us to overtake the problem of the extremely small percentage of geotagged tweets. In fact, the tweets we collect are automatically defined on a geographical basis, so no location extraction algorithm is needed, and all tweets are linked with spatial information. Moreover, the three methods we propose are extremely flexible, since they can be potentially applied to any spatial context and resolution (despite a certain degree of computational and coding effort being needed). Furthermore, the data that can be obtained by using our methods can contribute in constructing almost real-time indicators to support empirical researches or decision-making processes in different fields and contexts. Finally, the produced dataset can even be used as a benchmark for other geolocating algorithms, since data are fully provided with a reliable geographical information.

The three methods presented and compared in this paper rely on the "theory of circles" [49] (Section 4.1), which attempts to cover a set of territorial sub-areas by dividing each of them in circles. Then, we collect all available tweets sent within these circles, in order to obtain an as-much-complete as possible geographical coverage. More into details, Method 1 (M1, from here on) divides the sub-areas in three types of zones: NUTS not including big cities, NUTS including big cities, and NUTS including the capital city, London. Method 2 (M2, from here on) relies on circles whose radii are inversely proportional to a proxy of the sub-area population density. Method 3 (M3, from here on) is based on a high number of relatively small and equally sized circles covering the different NUTS. For a detailed description of the three methods, see Section 4. Our main research question is linked to the comparison of these three methods: Which method between M1, M2, and M3 represents the best option for collecting and geolocating tweets on a large

geographical area such as GB? In other terms: Is it more worthwhile to use a high resolution (but computationally "heavy") method like M3, based on a high number of small circles, rather than a simpler one (e.g., few and big circles, as in M1)?

In particular, the main research hypothesis of this work presumes that a balanced method such as M2 represents the optimal solution for the data collection. In order to test this hypothesis, the three methods are evaluated on the basis of the following four criteria: (a) Collection times required by each method for the data gathering, (b) the "size" of the collected information and the share of usable information, (c) the effort necessary for the pre-processing phase (i.e., when data are prepared for further advanced or more specific analyses), and (d) the accuracy of the geographic distribution reproduced by the collected tweets. These criteria are further discussed in Sections 4 and 5.

Our paper represents a noticeable improvement in comparison to the current literature. First, it proposes a collection method that takes full advantage of the potentialities of data retrieved from the web: We can produce, for relatively low costs, massive datasets, continuously and constantly updated, that provide a full coverage of a target area. Second, our method is able to retrieve information that is already fully geolocated. This reduces the problems related to the scarcity of location information. Moreover, it makes it possible to avoid the use of location extraction techniques that ask for long pre-processing times and frequently lead to not-completely reliable geographical information. In addition, the method we propose works on a semi-automatic basis (once it is set) and is characterized by a high level of flexibility (the researcher can adapt it to any country/context, by setting the desired geographical resolution). Data collected in such a way can then be used for any specific scope or purpose, in a wide range of applications, in order to support efficient and timely decision-making processes. Once such a potentially huge amount of data is stored, it remains available for being cleaned and/or filtered according to different criteria. This allows us to personalize and adapt the data to any scope, topic, and target, trying to maximize the potential nested in such a huge big-data mine.

## 4. Data Collection, Geolocation Methods and Evaluation Metrics

In this section, we first explain how we retrieved the data used for this research (Section 4.1) and then the three methods (M1, M2, and M3) we implemented in order to fully geolocate tweets in the GB NUTS macro-regions (Section 4.2). Finally, in Section 4.3 we introduce the metrics we used in order to compare the three proposed methods.

### 4.1. Data Collection

The data collection was based on the use of the search_tweets function of the rtweet R library. This function gathers tweets for a specified time interval (daily, in our case, i.e., from 0:00 a.m. to 11:59 p.m.) and for a specific geographical circular area (a "circle"). Circles are defined by three geographical variables (the radius of the circle, the latitude and longitude of the center). We developed the R code to automatically download the available tweets through the official Twitter API (https://help.twitter.com/en/rules-and-policies/twitter-api) for a given set of circles defined according to the three methods (M1, M2, and M3). The data were collected, in parallel, on three similarly featured desktop computers and individual API accounts (one for each method) from 15th January, 2019 to 15th February, 2019 (32 days). Despite the stable and high speed internet connection used, whenever a failure in the collection occurred, the process was able to automatically recover the lost information as soon as the problem was fixed. This allowed us to obtain a complete temporal coverage for the considered period.

In total, 119.5 million tweets were collected, for a cumulative time of about 749.3 h, leading to a total file size of 230 GB. For each tweet, the text and 88 other variables are available, providing details about the user that sent the post (e.g., the number of followers), about the data generation process (e.g., the type of used device), and about attached media files (e.g., URL to photos or videos, if available). In this article, the data is analyzed in an aggregate form, respecting EU General Data Protection Regulation (GDPR).

### 4.2. The Theory of Circles and the Coverage Problem

The data collection is based on the "theory of circles" [49], a method that has two aims: (i) To cover the sub-areas of interest as accurately as possible (in our case the eleven NUTS macro-regions of GB) by means of a set of circles (differently or equally sized); and (ii) to keep the setup of the method as simple as possible. Figure 1 (left) shows the GB NUTS regions we attempt to cover, whereas Figure 1 (right) displays how the "theory of circles" was applied in its first version to the GB case study.

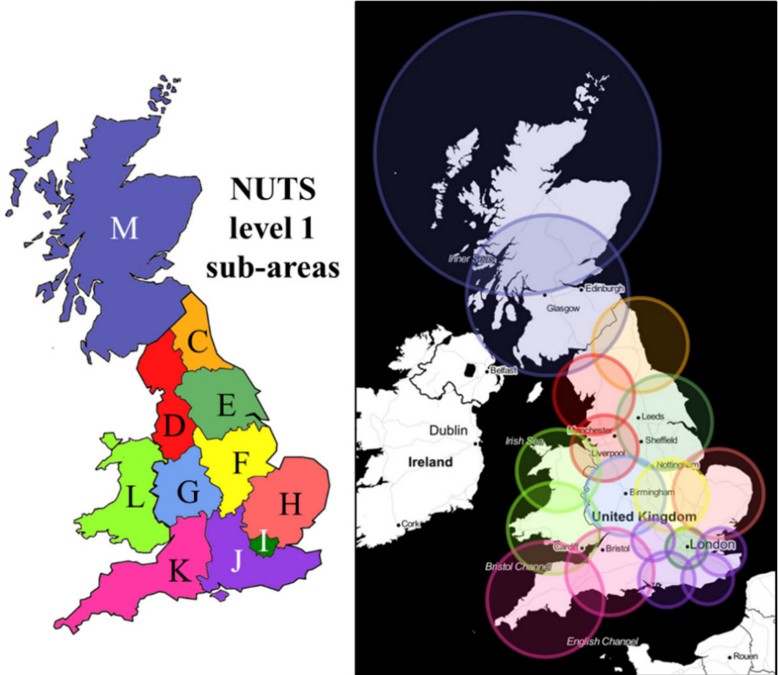

**Figure 1. Left**: Great Britain (GB) NUTS sub-areas; **right**: the original "theory of circles" applied to GB. Note. C = North East; D = North West; E = Yorkshire and the Humber; F = East Midlands; G = West Midlands; H = East of England; I = London; J = South East; K = South West; L = Wales; M = Scotland.

For the first attempt, both the size and the center of the circles were manually set, in order to fulfil the two criteria mentioned above. Initial attempts revealed that this approach did not lead to the desired result. More precisely, we identified three main critical issues: First, if circles were used sparingly, it was not possible to completely cover the entire area of some NUTS without frequently trespassing the NUTS borders (leading to the collection of a lot of duplicate tweets between the NUTS); second, a sparing use of circles led to highly overlapping circles within the NUTS (with the consequence of having a lot of duplicate tweets); third, especially for large circles and for circles including large cities, the collection of tweets was delayed and even frequently led to system crashes. In particular, the analysis of the tweets containing geotags has revealed that, if a circle is drawn too large, tweets at the edges of the circle may not be collected (depending on the number of tweets available for that specific area). This phenomenon, which we will refer to as the "circle coverage problem", is shown by the example displayed in Figure 2, which takes into account two circles.

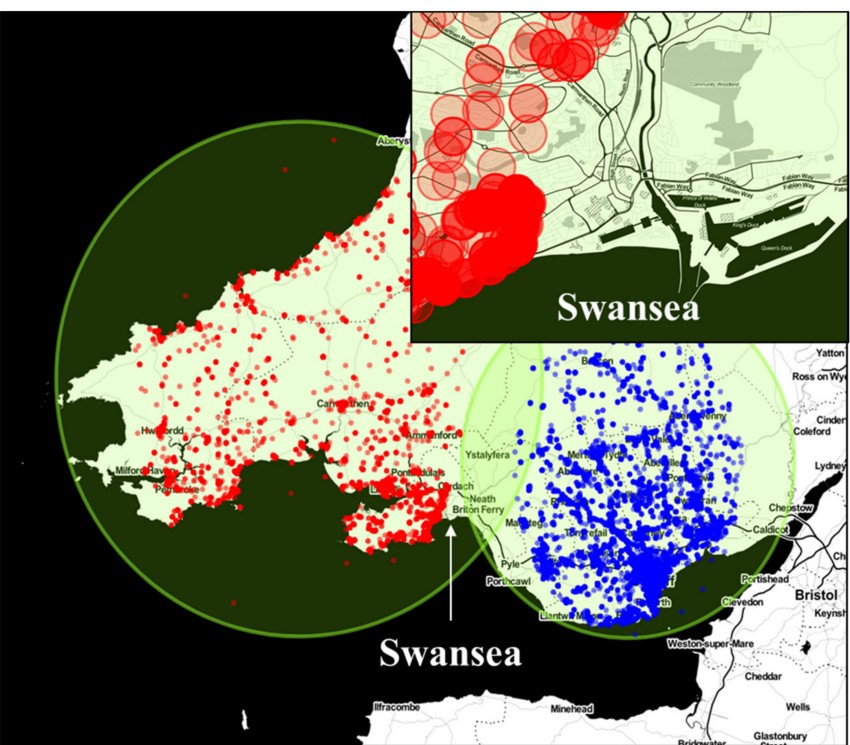

**Figure 2.** The "circle coverage problem".

If we consider the geographical distribution of the tweets with GPS coordinates (red marked points in Figure 2), collected for the left circle (that is large, in comparison to the population density), we notice that no tweets were collected for the area of the Swansea city center (zoomed figure in the top-right corner of Figure 2). The same phenomenon is also observed for the tweets marked in blue in the right circle of Figure 2: Tweets are only collected for the circle central area, leaving an empty ellipse between the two circles. It seems very unlikely that no tweets were sent in the intersection area (that includes the Swansea city center area). Furthermore, if the centers of the circles had been moved, posts would have also been collected in the ellipse area (and, of course, in the Swansea city center). This means that not all the available tweets are collected, if the radius of the circle is set too large (especially in a highly populated area). These findings led us to establish three possible alternative and optimized approaches, which might be able to overtake these issues.

The first method, M1, (shown in Figure 3, left) relies on the original method (see Figure 1, right) and additionally takes the "circle coverage problem" into account. More-over, with this method we aimed at reducing as much as possible both the overlaps between circles and their number. M1 is based on three main groups of zones: (a) London (divided in 73 relatively small circles, with an equally sized radius of 5 km); (b) big cities (i.e., cities with more than 350,000 habitants http://www.ukcities.co.uk/populations/), each represented by a separate circle which covers the entire urban area; and (c) other areas (i.e., areas not including London or big cities). For these areas, the centers and the radii of the circles have been defined in such a way that they reproduce as precisely as possible both the boundaries of the NUTS regions and the boundaries of the large cities. Following these criteria, for M1 we set a total of 246 circles with an average radius of 16.6 km.

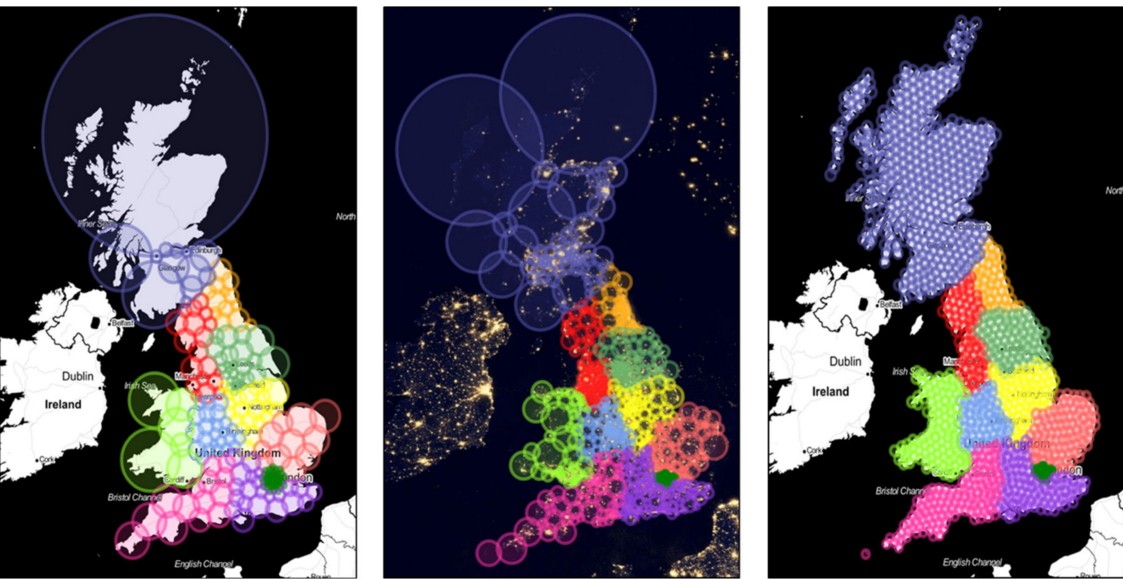

**Figure 3.** The "theory of circles" applied according to M1 (**left**), M2 (**center**), and M3 (**right**).

The second method, M2 (shown in Figure 3, center), takes the population density into account. This approach is based on the hypothesis that the amount of sent tweets should be higher in more densely populated areas than in less densely populated ones. Moreover, M2 was developed in order to face the "circle coverage problem" highlighted by Figure 2. Thus, tweets from more densely populated areas are collected by setting smaller circles, whereas larger circles could be used in areas characterized by a lower density, in order to keep the total number of circles low. Since the population density is difficult to determine for small areas (e.g., at community level), we use as a proxy the level of light emission observed in a photo layer from the "Earth at Night 2012" NASA (National Aeronautics and Space Administration) project (https://svs.gsfc.nasa.gov/30028). In particular, we employed the intensity of emitted lights at night shown by this image in order to set both radii and positions of the circles. This approach relies on the assumption that the population density strongly correlates with the intensity of the emitted light. We have subdivided the different NUTS sub-areas into small equal sized areas and categorized them depending on the emitted light intensity of the entire areas into three classes: High, moderate, and low light emitting areas. Next, the circles were set starting from the high light emitting areas to the less illuminated areas. In particular, we used radii of 5 km for high light emitting areas, radii of 10 km for moderate, and radii of more than 10 km for low light emitting areas (3 km radii were used for the NUTS I (London), as a consequence of the very high density observed in that area). Similarly to what has been done with M1, we attempted to map the NUTS sub-areas as precisely as possible by setting a proper positioning of circles' centers. In total, for M2 we used 847 circles with an average radius of 10.1 km.

The third method, M3 (see Figure 3, right), has the scope of minimizing the overlapping between the sub-areas (NUTS), in comparison to the two previous methods. Moreover, it aims at establishing an easy setup that could be fully, automatically, and more easily transferred to other countries. In particular, the centers of the circles were defined with the main aim of reducing as much as possible the circles overlapping between and within the NUTS regions. In order to achieve this objective, we set up a regular grid to cover the different NUTS sub-areas with 1257 relatively small and equally sized circles with radii of 10 km in order to ensure that the circles can be automatically positioned by a simple algorithm (3 km radii were used for the NUTS I (London)).

Note that, in order to completely cover the geographic area, for all the methods, a certain level of overlap between circles is necessary, which depends upon the size and the number of circles.

Despite being set for GB, these three methods of tweets collection are extremely flexible and can be easily adapted to other countries or contexts and also to other levels of geographical resolution. Each method has relative advantages and drawbacks and is linked to a certain level of effort and degrees of arbitrary in order to be set.

*4.3. Evaluation Metrics*

In order to evaluate and compare the three alternative methods introduced above, we take into account some factors, listed in the following.

As a preliminary step of the analysis, we study the geotagged tweets, i.e., tweets that contain coordinates. In particular, we aimed at ensuring that the tweets collected for a specific area through our methods also belong to the same area according to the GPS coordinates. If this is the case without exception, it can be assumed that the non-geotagged retrieved tweets are also correctly classified by our three methods within its corresponding geographical region.

After this preliminary phase, we compare the three methods according to the following criteria: (a) Collection times: We evaluate the collection times needed by each method in order to retrieve the total amount of tweets (see Section 5.1). (b) Dataset sizes: We analyze the "quantity of information" and "usable information" obtained by using each method (in terms of number of tweets and megabytes; see Section 5.2). (c) Pre-processing phase load: We evaluate the computational effort needed to pre-process the collected tweets, taking into account the share of duplicate tweets retrieved for overlapped circles (see Section 5.2). (d) Geographic distribution: We evaluate the relative level of territorial distribution comparing the percentage of collected tweets obtained by each collection method with the percentage of population by NUTS (see Section 5.3).

## 5. Analysis of Results

Our preliminary analysis focused on the tweets containing GPS coordinates, i.e., about 1% of the total number of collected tweets (about 387,840 for M1, 435,310 for M2, and 397,950 for M3; this percentage is equal to 1.03% for M1, to 1.01% for M2, and to 1.05% for M3). Comparing the geotag with the circles assigned by our methods, it resulted that all these tweets were correctly geolocated. Thus, we can also presume that all the non-geotagged tweets were actually sent from the assigned circle. Consequently, we did not expect any classification error for the tweets in the studied sub-areas.

*5.1. Collection Times*

A first criterion to evaluate the three alternative methods is the total time (daily average) needed by each of them in order to collect the total amount of tweets. This criterion was chosen to shed light on the relative efficiency of the data collection phase applying the alternative methods. The lowest daily average time is observed for M1 (7.52 h), whereas average retrieval times are longer for M3 (7.65 h, i.e., +1.7%) and for M2 (8.25 h, i.e., +9.7%).

Nevertheless, as we will show later, M2 is able to collect more tweets than M1 and M3, thus we should reason in terms of a more standardized measure, for example considering the time (in minutes) used to collect 10,000 tweets. According to this second indicator, more linked to the efficiency of the method, the best performance is observed for M2 (3.93 min to collect 10,000 tweets) in comparison to M1 (3.97 min; +4%) and to M3 (4.26 min; +8.4%).

If we perform the same analysis at the sub-area level, the situation is quite different: For just two out of 11 NUTS, the shortest time (in minutes) is obtained by M2, whereas the best performance is usually obtained by M1 (in 7 out of 11 NUTS). Of course, these results are a direct consequence of the mechanism behind the collection methods. M1 is the quickest method because it is generally based on bigger circles than the other methods (see Figure 3, left). However, this unfortunately comes with a relevant drawback: In marginal areas of such big circles, mostly in densely populated regions, we noticed an abundance of under-coverage or even missing coverage (as a consequence of the "circle coverage problem" mentioned in Section 4.2). This missing coverage, at the same time,

is a possible reason for the faster collection (we are able to collect less tweets, and times are shorter). The results obtained using M3 are not unexpected, since this method uses the highest number of (small) circles. On the one hand, this attenuates the "circle coverage problem" and allows for a more precise "reproduction" of NUTS borders; on the other hand, the presence of such a big number of circles makes the retrieval process slower and asks for longer processing times. Between the two extremes represented by M1 and M3, M2 seems to be a very good compromise, fulfilling both the need for fully covering sub-areas and the need for reducing the collection times.

### 5.2. Dataset Sizes and Non-Duplicate Tweets

A second criterion of comparison is the total amount of data that the three methods were able to collect over the whole collection period. This criterion is chosen in order to assess the relative capability of obtaining the most complete possible collection of available tweets, of which exact amount is unknown. For the purpose of simplicity, the data size is specified in megabytes (MB). M1 was able to produce a dataset with a size of 8189 MB, in comparison to 8025 MB obtained with M2 (−2%) and to 6815 MB obtained by M3 (−16.8%). However, at the NUTS level, the results are not completely consistent: In fact, we notice that M1 and M2 show the highest size for five NUTS each, whereas M3 has a higher value just for one NUTS out of 11.

These figures do not take into account the relative efficiency: For this reason, it is better to make the comparison in terms of number of MB or number of tweets collected per hour. From the first point of view, M1 is still the most preferable method, being able to provide a total amount of 34.0 MB per hour, whereas both M2 (30.4 MB) and M3 (27.8 MB) perform considerably worse.

Nevertheless, so far we have just taken into account the raw data (i.e., the total amount of produced megabytes), without considering the amount of tweets and the issue of overlaps between circles and of duplicate tweets (see Section 5.2). Considering these aspects, the method that performs better is M2. This last approach provided 163,182 tweets per hour, whereas by means of M1 and M3 we collected 160,043 and 154,928 tweets per hour, respectively. Moreover M2 is able to collect 157,079 unique tweets (i.e., excluding duplicates) per hour, versus 151,634 unique tweets collected by M1 and 141,152 collected by M3. Thus, M2 seems to be the most efficient method in collecting useful and usable data, per time unit.

### 5.3. Pre-Processing Phase Load

In this work, we attempt to cover a geographical area (GB) using circles. However, if one aims at fully covering a whole area using a certain spatial resolution (e.g., studying sub-areas such as GB NUTS), a certain level of overlapping between circles should be expected (see Figure 3, left and center, or Figure 2, where the phenomenon is more evident). This overlap could cause duplicates, because we can retrieve the same tweet twice (e.g., for both the overlapping circles of Figure 2), three times (if there is an overlap of three circles; see, for example, the intersection near Cardiff in Figure 1 (right), and so forth. Consequently, after the daily data collection is completed, it is necessary to remove the duplicate tweets collected in overlapping circles. For this purpose, we identify duplicates by means of the unique code of each tweet; then, over this group of duplicates, we keep just one unique tweet that is randomly reassigned to one of the overlapping circles.

We expect to obtain a high percentage of duplicate tweets when we have a lot of circles or by setting big circles. This would imply a longer pre-processing phase to identify and randomly reassign unique tweets. Thus, both the average size and the number of circles directly affect the computational effort necessary to obtain high quality data. As a consequence, the percentage of duplicate tweets can be used as a measure of computational load; the best method of collection should minimize this percentage by lowering both the overlap of circles and the percentage of duplicates. This also means that the best method should also maximize the percentage of unique tweets.

Furthermore, there are two types of overlap: Within-NUTS, when the overlap is between two (or more) circles within one specific NUT; and between-NUTS, when two circles overlap across the borders between two (or more) NUTS. In our case, we are mostly interested in reducing the between-NUTS overlap. In fact, the latter is more dangerous, being a potential source of location bias, when we randomly reassign an unique tweet to the "wrong" NUTS. Thus, the best method of collection should also (mostly) reduce the percentage of between-NUTS overlap, in order to reduce this potential bias.

In Table 1, the main measures linked to the pre-processing phase load are reported. Contrarily to what we expected, the bigger the circles, the higher the rate of unique tweets: In fact, M1 reaches the highest percentage of unique tweets (94.73%; i.e., just 5.27% of tweets are selected where circles overlap). This result could be explained, at least in part, by two factors. First, by the "circle coverage problem" introduced in Section 4.2: If the number of tweets in a circle is huge (and this is more likely when the circle is big), the algorithm is not able to completely collect tweets sent in that area, mostly in the marginal parts of the circle, where an overlap with other circles is more likely. For smaller circles, this problem attenuates, leading to a higher percentage of unique tweets. Second, if circles are relatively small or very small (e.g., in M3), more circles are necessary to fully cover a country (and its sub-areas), and more frequently we obtain overlaps, increasing the probability of obtaining duplicates. However, if M1 seems the best method in terms of percentage of unique tweets, the shares of tweets overlapping within and between NUTS tell a different story.

**Table 1.** Unique tweet and duplicates by method of collection (percentage values; the best performance measures are in bold).

| Method | Unique Tweets | Duplicates | | |
|---|---|---|---|---|
| | | Within NUTS | Between NUTS | Total |
| M1 | **94.73** | 4.98 | 0.29 | 100 |
| M2 | 93.97 | **3.51** | 2.52 | 100 |
| M3 | 91.74 | 8.16 | **0.10** | 100 |
| Avg. (unweighted) | 93.48 | 5.55 | 0.97 | 100 |

In terms of within-NUTS overlaps (third column of Table 1) the best method is M2 (3.51% of tweets overlapping within-NUTS, in comparison to 4.98% and 8.16% obtained with M1 and M3, respectively). Thus, M2 is the best option: It is a good compromise between the arbitrary definition of the circles used in M1 (for this method, bigger circles cause more overlap within NUTS, see Section 4.2) and the big number of circles asked by M3 (that shows the worst performance, since a lot of small circles are used to cover sub-areas).

The lowest between-NUTS overlap is registered for M3 (0.1%, fourth column of Table 1). The use of several small circles allows us to more precisely reproduce sub-areas borders (reducing considerably the between-NUTS overlap). M1 also performs nicely in terms of between-NUTS overlap (0.29%), whereas M2 reaches the highest percentage (2.52%). These results can guide a further development of the current methodology: M2 can probably be further refined using relatively smaller circles in the area closer to the NUTS borders, in order to reduce the percentage of between-NUTS duplicates.

If these are the general results over GB, the percentages of unique and duplicates within and between NUTS seems to vary considerably by geographical sub-areas, as shown in Figure 4.

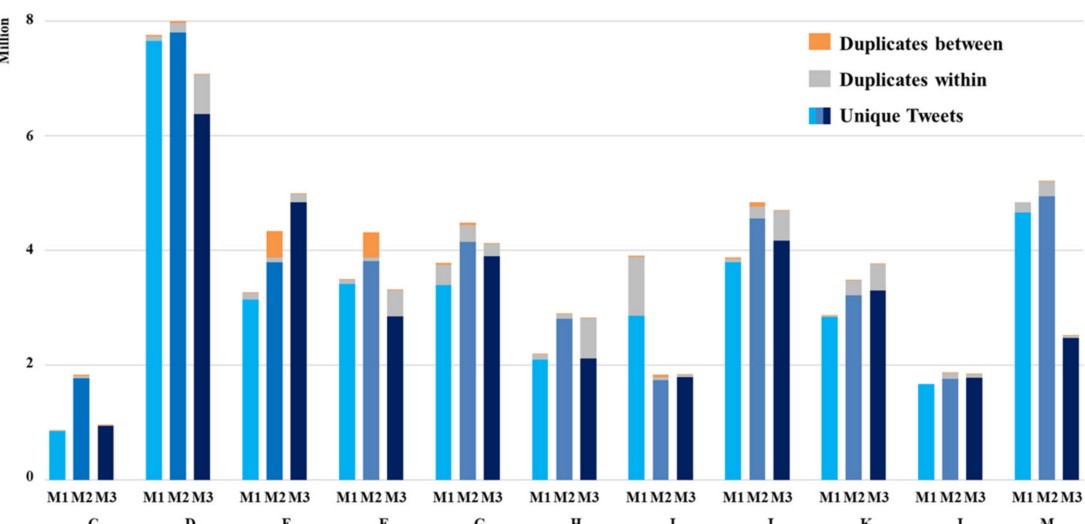

**Figure 4.** Number (in million) of unique and duplicate tweets (by NUTS and by method).

If we consider the distribution by NUTS, M2 collected the highest number of tweets in 8 out of 11 NUTS. In total, with M2 we were able to obtain 45,861,979 tweets, i.e., a quantity that is +12.9% and +10.9% higher, respectively, in comparison to 40,627,058 tweets collected using M1 and to 41,336,142 using M3. Moreover, M2 also shows the best performances in terms of unique tweets in 7 out of 11 NUTS, while performing similarly to M3 for NUTS L. The total number of unique tweets collected by M2 is 43,096,363, +12.0% in comparison to 38,487,675 unique tweets retrieved with M1 and +13.6% in comparison to 37,921,166 collected by means of M3. In addition, for some NUTS, M1 and M3 highlight relevant problems of within-NUTS overlap: Using M1, we obtain the 26.33% of duplicates for NUTS I, whereas with M3, the same percentage is equal to 9.77% for NUTS D, to 14.07% for F, to 24.83% for H, to 11.24% for J, and to 12.48% for K.

On its side, M2 shows high between-NUTS percentages (>10%) for NUTS E and F: This confirms what previously suggested by general results, i.e., that M2 should undergo a further revision of the procedure used to set circles, mostly for zones close to the NUTS borders and for specific NUTS such as E and F.

*5.4. Geographic Distribution*

As a further criterion for evaluating the three collection methods, we measure their relative capability in reproducing, with the collected tweets, the population distribution across NUTS. In particular, in highly populated areas we expect to collect a proportionally higher percentage of tweets. Thus, we evaluate the difference between the percentage of tweets collected in each sub-area on the total number of collected tweets, and the percentage of population living in the same area on the total GB population (reference year: 2018[15]). This method of evaluation is not perfect, because it assumes that the distribution of the population by age, gender, education level and, mostly, by Twitter users and type/frequency of usage is similar in the different NUTS, and we can be quite sure it is not. Nevertheless, if the scope is to compare our three alternative methods of collection, we believe that these differences can be considered negligible and do not considerably affect the comparison and the relevance of our results. Thus, we believe this is a reliable criterion in order to evaluate the quality of the territorial coverage of the three methods.

Considering the whole GB, the best performance is obtained by M2, that shows the lowest absolute average difference between the distribution of tweets and of the population (2.95%). Both M1 (2.99%) and M3 (3.06%) perform worse. Figure 5 shows the detailed results, in terms of differences between tweets and population percentages by NUTS.

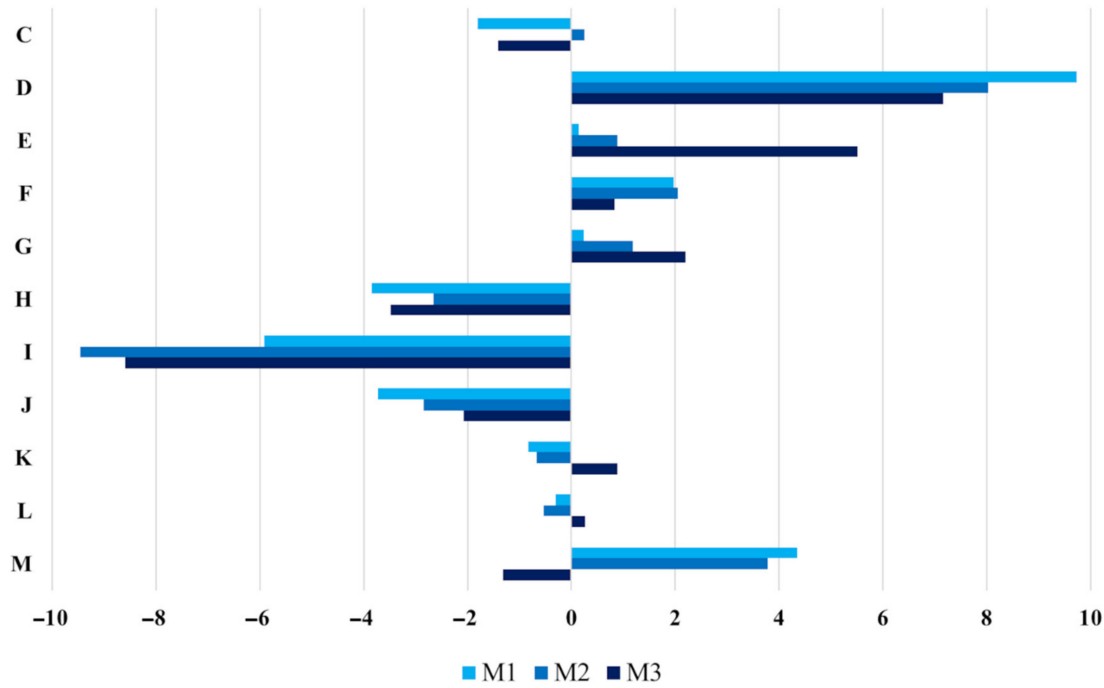

**Figure 5.** Differences between percentage of tweets and percentage of population (by NUTS and by method).

In Figure 5, we notice that the relative performance of the three methods is varying a lot at the local level. For three NUTS out of 11 (C, H, K), the best performance is obtained by M2, whereas the lowest difference is obtained by M1 in 3 NUTS (E, G and I) and by M3 in the remaining 5 NUTS (e.g., D or F). In Figure 5, we also observe that for NUTS D, E, F, and G (that correspond to East and West Midlands, Yorkshire, and North West) all three methods show positive differences: This probably means that the population of these sub-areas is more active in posting tweets. On the contrary, the negative values obtained with all methods for NUTS H, I, and J (corresponding to East of England, London, and South East) could be probably caused by two reasons. On the one hand, those areas, which show very high levels of population percentage, could be affected by the "circle coverage problem". On the other hand, the population living in such areas could be less active in sending posts (for example because it is more heterogeneous, in terms of some socio-demographic variables, and it includes groups of people that are less inclined to use Twitter or, in general, social media).

## 6. Discussion

### 6.1. Comparison of the Different Methods

The data collected by means of our three methods have three main relevant advantages, in comparison to what it is usually done. First, data are automatically and exhaustively linked to a specific sub-area; thus, the implementation of algorithms in order to geolocate tweets (and the linked potential bias) can be avoided [12,20,42,44,46]. Second, the methods of collection are extremely flexible and can be both personalized for any geographical area and adapted to any level of territorial detail. They allow us to completely cover any country and the sub-areas of interest collecting all tweets available through the Twitter API. Third, the wide selection of tweets produces a dataset that can be immediately used for any purpose (from sentiment analysis, to estimation or update of indicators of any kind, from longitudinal analyses of a wide range of contexts, to spatial studies of socio-demographic phenomena).

Nevertheless, which is the method, among the three considered, that allows us to optimize the process of tweet retrieval? Generally speaking, the best choice is M2. And this holds for various reasons. First of all, because it is a good compromise between M1

(which is a method based on big circles quite arbitrary and relatively easy to be set and implemented) and M3 (that is rather based on several small circles that share a territorial area in very small circular zones). M2 is not-so-arbitrary (being the size of circles set according to a proxy of the population density) and not-so-detailed (providing anyway a complete coverage, in terms of collected data). Table 2 summarizes our results comparing the three methods according the main variables we evaluated.

**Table 2.** Number of used circles, total amount of tweets, and collection time (best performances are in bold).

| Method | M1 | M2 | M3 |
|---|---|---|---|
| Used circles (no.) | **246** | 847 | 1257 |
| Total amount of tweets (million) | 38.4 | **43.1** | 37.9 |
| Overlap within NUTS (%) | 5.3 | **3.7** | 8.9 |
| Overlap between NUTS (%) | 0.3 | 2.7 | **0.2** |
| Unique tweets (million) | 36.4 | **40.3** | 34.5 |
| Collection time (per 10,000 unique tweets in min) | 4.0 | 3.9 | **4.3** |

M1 is easy to set up, given the low number of circles (246) that are used (see Table 2). For M2, 847 circles were adapted to the population density, while M3 relies on 1257 small circles of the same size. M2 is characterized by the highest total amount of collected tweets compared to the other two methods. This aspect is also confirmed in terms of the number of unique tweets. Likewise, M2 produces the lowest proportion of overlap within NUTS, where M3 produces the least amount of overlap between NUTS. Considering the time needed to collect the tweets, M2 appears to be the most efficient method.

However, the comparison between our methods is not as easy as it seems, and the conclusion is not-so-direct, mostly if we consider the performance of the three methods at the sub-area level; moreover, our results show that there are still clear margins to improve even our current best choice (M2).

In general, we found that M1 is able to collect the biggest quantity of data (in terms of produced megabytes). Nevertheless, considering sub-areas we found that M2 is the method that produces more data for 5 out of 11 NUTS. Moreover, M2 allows us to maximize both the total amount of tweets collected per hour and (most of all) the amount of unique tweets collected per hour. This is an important competitive advantage. On the one hand, M2 maximizes the amount of "usable" information (retrieved by time unit) and reduces the number of tweet duplicates as well as the potential biases caused by the reallocation of tweets corresponding to overlaps between circles. On the other hand, M2 also allows us to reduce the pre-processing times, since both the amount of duplicates and of tweets to be reassigned are reduced. Thus, M2 results in being not just a good compromise to be implemented, but also, relatively to M1 and M3, the most efficient method in terms of amount of information collected and processing times.

Furthermore, even in terms of total collection times, M2 allows us to retrieve the highest number of tweets and unique tweets: The coverage of the whole country and of sub-areas are generally optimized, using this method. Actually, M1 results as the quickest method at the local level (i.e., in most of NUTS), in terms of raw data, but this method leads to a higher percentage of areas with missing tweets, mostly due to the "circle coverage problem" (see Section 4.2). M3 is, instead, the least performing method in both terms (globally and locally), mainly due to the extreme level of detail that asks for longer computational times, a higher effort to be set, and a higher pre-processing phase load. Thus, even in terms of collection times, M2 seems to be a very good compromise and an improvement, in comparison to the two extremes represented by M1 and M3.

*6.2. The Implication for Open Innovation*

Based on current empirical evidence on the adoption of open innovation, this concept can be considered a global trend occurring in almost all industries and markets [50], globally.

Open Innovation is not a clear-cut concept and can take many forms [51], and the definitions used can differ significantly. However, most have in common that the goal of Open Innovation is both the integration of external sources of innovation into the firm and the identification of external pathways for the commercialization of internally sourced innovations [52]. In this framework, and mostly from the first perspective, the implementation, the study of advantages and drawbacks/limitations, and the use of new types of data sources such as Twitter can potentially play a very important role. The central point of the basic assumption of open innovation is that both the integration of external sources and the identification of external pathways are profitable for the actors [52]. Schroll and Mild [50], after a meta-analysis of scientific articles regarding open innovation, described the definition of Chesbrough [53] as a very general and comprehensive definition: "Open innovation is the use of purposive inflows and outflows of knowledge to accelerate internal innovation, and expand the markets for external use of innovation, respectively. This paradigm assumes that firms can and should use external ideas as well as internal ideas, and internal and external paths to market, as they look to advance their technology". Even from this perspective, social media such as Twitter can play a relevant role (for example in integrating more traditional sources of data) as both newer (and maybe more effective) ways to retrieve information and to serve more commercialization purposes. However, this novelty needs to be further studied, in order to fully detect limitations, potentialities, and how to use these sources in a specific field.

In the literature, two central directions of open innovation are mentioned, Inside-Out Open Innovation (cases in which innovations made within an actor are, for example, licensed) and Outside-In Open Innovation (cases in which an actor provides innovations for external patents) [54,55]. According to Schroll and Mild [50], the inbound mode is much more widespread than the outbound mode and was already used in 2011.

We are convinced that the method we have presented is suitable for a collection of geolocated tweets that could be applied for both directions of open innovation. In times of "market uncertainty" with rapidly changing customer needs and other market-related effects, companies feel compelled to react flexibly to these changes [56]. Thus, for instance, not only a content and quantitative analysis of posts from social networks (e.g., tweets) could help to evaluate a marketing campaign, but also a temporal and a spatial component would provide important information that could integrate or even, in the future, substitute more traditional data sources.

Clusters and regional innovation systems are important for Open Innovation because the flow of knowledge between actors is crucial in this framework. An optimal open innovation strategy can benefit from multiple linkages to different institutions to increase the efficiency of open innovation itself in different regions or nations [57]. Moreover, sectors with many actors are those which could mostly benefit from open innovation, as opposed to those dominated by monopolists [58]. The method we present could be an enrichment mostly for a network of actors such as these, including research, public institutions, and businesses with regard to application to a wide range of areas already introduced in previous chapters (disaster management, public opinion, etc.). This is especially true if data from different regions and countries are provided by different actors and combined, together, in an integrated way and with a consistent and robust methodology, into a common data framework.

## 7. Conclusions

In this paper, we compare three different methods to retrieve tweets posted through Twitter. Although all three methods are based on the common "theory of circles", they are developed on the basis of different criteria (mainly related to the size and number of circles and, thus, to the coverage of the sub-areas). Using these three methods in parallel, we collected for one full month, starting from mid-January 2019, data all over GB. Our objective is to find the most efficient method of tweets collection, relying on empirical evidence on the collection process. This aspect is very important, because only data that are efficiently

and quickly collected and processed can be used for producing information that helps in developing research or in taking "solid" (supported by evidence) and timely decisions in almost every field or context.

From the point of view of these criteria, our results generally suggest that M2 is the best compromise among the methods we studied and compared. Nevertheless, as already anticipated, M2 is not a "perfect" method and needs to be further enhanced. For example, it has problems (in few NUTS, luckily) with high percentages of tweets overlapping between NUTS. These are critical cases, since they can cause bias in the random reassignment of unique tweets. Probably, for particular NUTS the method should be refined, using other criteria in addition to the population density when defining the circles' size (e.g., the proximity to the NUTS borders). This means that, for example, we could try setting circles with a radius inversely proportional to the population density, but also inversely proportional to the closeness to the borders of the sub-areas we aim at studying. On their side, both M1 and M3, which do not rely on criteria such as the population density, show relevant problems with the percentages of overlapping tweets, reaching particularly high levels for some NUTS (such as I and H, with more than 24% of overlapping tweets within NUTS).

Our work is currently affected by some limitations. These, nevertheless, can be ideas for further developments of this research. First of all, all our methods of collection (M2, mostly) are flexible, since one researcher can set the methods according to the requested spatial resolution. Nonetheless, they cannot be re-adapted a-posteriori (i.e., after the data are collected). For example, if one is interested in using a higher resolution, the dataset already produced cannot be downscaled, since the initial interest was for wider areas. This means that the level of territorial details has to be carefully planned, before starting the collection.

Moreover, the Twitter API allows us to retrieve a sample that corresponds to a maximum of 1% of all posted messages. Nevertheless, the methods used by Twitter to sample these tweets is currently unknown [59]. However, we believe that the limitation to 1% of the total number of tweets is not a factor that could affect the relevance or the implications of our results. One potential solution to overcome the 1% limitation is to use the Twitter Firehose that allows access to almost 100% of all public tweets. However, "a very substantial drawback of the Firehose data is the restrictive cost" [11] p. 1), and this strategy does not represent a "perfect" solution, anyway: Other researchers found that some publicly accessible tweets are missing from the Firehose [59]. Nonetheless, without a doubt, by facing these costs, one will be allowed to replicate our study or to conduct a deeper experiment on this same topic.

Another limitation of our study is linked to the data providers, i.e., to the population of Twitter users. These users simply do not represent 'all people', but are rather a particular sub-set of them. This means that Twitter users cannot be considered a representative sample of the global population, but they are characterized by different features. To make things worse, some users have multiple accounts or they are just 'listeners', some accounts can be used by multiple users or can be 'bots' [59]. However, such kinds of limits are actually referred to the datasets our methods allow to produce and affect all the existing research studies based on Twitter data. Thus, this does not reduce the validity and the importance of our findings (and mostly, the comparison between our methods).

Although we believe that our judgement is based on a complete set of valid indicators, our study could be further developed comparing other aspects, such as, for example, the density of tweets and the density of population observed in sub-areas or the time necessary to clean, classify, analyze, or develop more advanced tasks starting from the collected tweets (e.g., applying sentiment analysis methods).

Our work could be useful as a starting point for a wide range of research and applied research fields. For example, Cachia et al. [60] highlighted a link between social media and innovative activities in open innovation (in particular, in the communication processes).

In this regard, Zamarreño-Aramendia et al. [10] believe that the context in which social media can be used for open innovation still needs to be further studied.

From a more practical perspective, Howe [61] suggests to public administrations to use open innovation and corporate R&D to fix issues in the communication and prevention of natural disasters. In such a framework, the use of social networks (and of Twitter, especially) should enable public officials to collect "knowledge produced by different involved stakeholders, as well as to use all these knowledge synergies in order to establish mid- and long-term prevention public policies" [10] p. 13. Against the background of a natural disaster, the data collection and processing described in this paper could be reduced from a daily to few hourly intervals in order to be able to obtain timely information.

Another extremely relevant point is the evaluation of how it would be possible to integrate data retrieved from such a new kind of source with more traditional survey data or official statistics data. Would it be feasible? If so, to what extent and under which conditions? What would be the main challenges in realizing and optimizing this type of data integration in order to produce a new enhanced knowledge and, at the same time, to reduce existing quality and representativeness issues?

**Author Contributions:** Conceptualization, D.T., M.C. and S.S.; methodology, D.T., M.C. and S.S.; software, S.S.; validation, D.T., M.C. and S.S.; formal analysis, S.S.; investigation, D.T., M.C. and S.S.; resources, D.T., M.C. and S.S.; data curation, S.S.; writing—original draft preparation, D.T., M.C. and S.S.; writing—review and editing, D.T. and S.S.; visualization, S.S.; supervision, D.T., M.C. and S.S.; project administration, D.T., M.C. and S.S.; funding acquisition, D.T., M.C. and S.S. All authors have read and agreed to the published version of the manuscript.

**Funding:** This work was supported by the University of Bergamo [60% University Funds and project "STaRs (Supporting Talented Researchers)—Azione 3: Outgoing Visiting Professor 2019"]. We acknowledge support by the Open Access Publication Funds of the Göttingen University.

**Data Availability Statement:** Available on request.

**Conflicts of Interest:** The authors declare no conflict of interest.

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
