# Peer review of "Comparing Methods to Collect and Geolocate Tweets in Great Britain"

_2199-8531, doi:10.3390/joitmc7010044_

Round 1

Reviewer 1 Report

I read the study with great interest. The work was done scientifically competently, and in my opinion, it can be published in its current form.

However, I write a couple of not very essential remarks
- seems strange that the authors did not present in the literature review methods that are alternative to the theory of circles
- The authors describe in general the scope of application of the methods, however, for different purposes, different parameters may be more in demand. It would be interesting for me to read what the authors would suggest for an emergency that requires the prompt collection of information about what is happening in a certain location (for example, a flood) to coordinate the activities of the population, rescue services, etc.

Author Response

We thank the reviewer for their careful reading of the manuscript and their constructive remarks. We have taken the comments on board to improve and clarify the manuscript.

Reviewer 2 Report

Introduction section is too long. There is no clearly stated scientific problem. Comparison of data collection mehod is not scientifically relevant problem. Some part of introduction – mainly two last paragraphs, sounds like conclusions. In Introduction section the Authors should learly describe scienticic gap and mention main and specific goals of the study. Maybe: development and description of assessment methodology of SNS data collection method according to selected indicators. And there is a lack of clear scientific jusfification, why these indictors are relevant?

From lines 41 to 57 there is no quote. IMO this is not correct for a research paper. Lines 120-141 - reflect the conclusions rather than the introduction.

In conclusion: only when I read the 3rd section, it was clear for me what Authors want to achieve ?.

The literature review section is relevant and describes the theoretical framework of the problem well.

Section 4 should be tittled as Analysis of results.

The discussion part should be prepared after the results are presented. There is a short paragraph in the Conclusions (lines 587-597) in which the Authors try to discuss their results based on literature resources. Maybe it's a good starting point for setting up a real discussion section. In general, conclusions should not contain citations. Conclusions means that the Authors formulated final statements regarding the obtained results, without discussing them with other literature resources.

Point 6. Patents should be removed.

The references list is not in accordance with the requirements of the JOItmC.

Author Response

(The authors gave the same response as above.)

Reviewer 3 Report

Dear authors,

first of all, I want to congratulate you on your research. The paper is very well written and touches the essential parts for a paper to be considered for publication in a journal at this level. In my opinion, the paper "Comparing Methods to Collect and Geolocate Tweets" falls into the scope of the "Journal of Open Innovation: Technology, Market, and Complexity" and represents a very nice contribution.

A few very minor comments:

1. Line 63, you could omit the text (https://twitter.com/). It's already well known.
2. I would pay attention to the name of the chapter "6. Patents".
3. In my opinion, "Appendix A" could be omitted from the published version.
4. References should be formatted according to MDPI format.

Best regards and good luck with your research.

Author Response

(The authors gave the same response as above.)

Reviewer 4 Report

Before starting a more in-depth review, I would like to point out that there is no mention of the concept of open innovation. I recommend that you include a section in the theoretical framework in which you include the following quotes:

  • Yun, J.J.; Zhao, X.; Jung, K.; Yigitcanlar, T. The Culture for Open Innovation Dynamics Sustainability 2020, 12, 5076
  • Yun, J.J.; Zhao, X.; Park, K.; Shi, L. Sustainability Condition of Open Innovation: Dynamic Growth of Alibaba from SME to Large Enterprise. Sustainability 2020, 12, 4379
  • Zamarreño-Aramendia, G.; Cristòfol, F.J.; de-San-Eugenio-Vela, J.; Ginesta, X. Social-Media Analysis for Disaster Prevention: Forest Fire in Artenara and Valleseco, Canary Islands. J. Open Innov. Technol. Mark. Complex. 2020, 6, 169.

Although the objectives appear, I recommend that you unify them at the end of the introduction to facilitate reading. So that each of the previously explained objectives is recorded on one line.

There are parts of the introduction that should appear in the theoretical framework. The introduction should be shorter.

I recommend that at the end of the introduction you include the structure of the paper.

The structure of the paper should include: Introduction, theoretical background, materials and methods, results, dicussion and conclusions.

The discussion section of the results is not a real discussion. There is no link to the theoretical framework, but rather a reference to presenting the results. The presentation of the data must be clearly differentiated from the discussion.

Their conclusions are closer to a discussion. They must rearrange the ideas so that the paper is clearly defined.

In general the idea is good, the development is acceptable, but the order can be improved. It is a matter of form, not substance.

In any case, remember the name of the magazine to which you are submitting the paper. If the magazine is about open innovation, you should go deeper into this concept and propose a definition and a relationship of open innovation throughout the paper: from the abstract to the conclusions.

For example: in their introduction they talk about "decision-making processes", and there you can find the link to open innovation.

Go ahead, because the work is good, but it must be focused on the reality of the scope of the magazine.

Author Response

(The authors gave the same response as above.)

Round 2

Reviewer 2 Report

Dear Authors, in my opinion, the paper is now suitable for publication.

Author Response

Thank you for your positive feedback. We are pleased that we seem to have addressed all your helpful points and like to thank you again for your work.

Reviewer 4 Report

You have done an excellent job of adapting your paper to the structure. I think you have understood some of the basic elements of my review, such as the need to bring the paper closer to the magazine's aim&scope. Congratulations

Author Response

(The authors gave the same response as above.)
